# Pennycress as a Cash Cover-Crop: Improving the Sustainability of Sweet Corn Production Systems

**Sarah A. Moore** [1], **M. Scott Wells** [1,*], **Russ W. Gesch** [2], **Roger L. Becker** [1], **Carl J. Rosen** [3] and **Melissa L. Wilson** [3]

[1] Department of Agronomy and Plant Genetics, University of Minnesota, St. Paul, MN 55108-6028, USA; moor1579@umn.edu (S.A.M.); becke003@umn.edu (R.L.B.)

[2] United States Department of Agriculture, Agricultural Research Service, Morris, MN 56267, USA; Russ.Gesch@ars.usda.gov

[3] Department of Soil, Water, and Climate, University of Minnesota, St. Paul, MN 55108-6028, USA; rosen006@umn.edu (C.J.R.); mlw@umn.edu (M.L.W.)

* Correspondence: mswells@umn.edu

**Abstract:** Commercial sweet corn (*Zea mays* convar. *saccharata* var. *rugosa*) production has a proportionally high potential for nutrient loss to waterways, due to its high nitrogen (N) requirements and low N use efficiency. Cover crops planted after sweet corn can help ameliorate N lost from the field, but farmers are reluctant to utilize cover crops due to a lack of economic incentive. Pennycress (*Thlaspi arvense* L.) is a winter annual that can provide both economic and environmental benefits. Five N-rates (0, 65, 135, 135 split and 200) were applied pre-plant to sweet corn. After the sweet corn harvest, pennycress was planted into the sweet corn residue with two seeding methods and harvested for seed the following spring. Residual inorganic soil N ($N_{min}$), pennycress biomass, biomass N and yield were measured. The nitrogen rate and seeding method had no effect on pennycress yield, biomass, or biomass N content. The nitrogen rate positively affected $N_{min}$ at pennycress seeding, wherein 200N plots had 38–80% higher $N_{min}$ than 0N plots, but had no effect on $N_{min}$ at pennycress harvest. Control treatments without pennycress had an average of 27–42% greater $N_{min}$. In conclusion, pennycress can act as an effective N catch crop, and produce an adequate seed yield after sweet corn without the need for supplemental fertilization.

**Keywords:** *Thlaspi arvense* L.; *Zea mays* convar. *saccharata* var. *rugosa*; nitrogen management; cover crop

## 1. Introduction

Sweet corn (*Zea mays* convar. *saccharata* var. *rugosa*) is grown in many regions across the world, including the Americas, Asia, Europe, Middle East, Africa and Australia [1–3], with the highest production in the United States. The upper Midwest USA leads the nation in production, accounting for 37% of the total sweet corn hectarage in the United States [4]. Even so, sweet corn makes up less than 1% of total cropland in the upper Midwest, while maize and soybean together account for 71% [5]. However, sweet corn has a proportionally larger potential to contribute to nitrogen (N) pollution, with three times the residual inorganic soil N of grain maize and soybean. This occurs as a result of the high fertilizer requirements to optimize ear or cut-kernel yield, which subsequently results in high N residue left on the field after harvest [6–8]. Sweet corn requires approximately 200 kg N ha$^{-1}$ to achieve high-quality ears [6,9]. While this fertilizer rate is comparable to recommendations for field corn (*Zea mays* L.), sweet corn is seeded at lower population densities and is harvested as a fresh vegetable rather than at grain maturity, meaning that 34.3% to 50% of the N applied as fertilizer is not utilized by the crop, and is susceptible to transport off-site [5,10,11].

Residual inorganic N has the potential to contaminate surface and groundwater, causing negative effects on both human and environmental health [12]. In humid temperate climates such as the upper Midwest USA, the majority of N leaching occurs during the fall and spring due to rainfall and snowmelt during the shoulder season in annual summer cropping systems [13]. Throughout much of the upper Midwest, the land is left fallow from November through May, exacerbating N loss. One way to mitigate the issue of N loss is through the use of winter annual cover crops.

Winter annual cover crops are cold-hardy, and thus can be temporally positioned to uptake N in both the fall and the spring when the risk of leaching is greatest [14–16]. Compared to bare soil, winter annual cover crops can reduce N leaching by up to 80% mg L$^{-1}$ [17], which is twice that of autumn cover crops [18]. The dominance of winter-active habit cover crops in their ability to reduce N loss both within and across species has been corroborated in a number of studies [14,19–22].

However, cover crops are only utilized on 2.3% of cropland in the Midwest USA [23]. Research conducted in the state of Iowa shows that producers stated more economic incentives and diverse markets are necessary to increase cover crop adoption [24]. Other studies have similarly listed the need for better facilitating infrastructure, such as technical assistance [25] or cost-sharing [26], in order to increase the use of cover crops. The Chesapeake Bay area in the USA is one example where the implementation of legislation that provides a monetary reward to farmers that use cover crops has been a success. The Maryland Department of Agriculture's Cover Crop Program provides grants to farmers who plant cover crops, and from 2016 to 2017, over 226,210 ha of cover crops were planted in the state [27], which is equal to 43% of total commodity hectarage, the highest in the nation [28]. Another alternative to legislative incentives is on-farm revenue increases through the deployment of "cash cover crops" [29,30]. Cash cover crops are crops specifically designed to provide all the same environmental benefits as a cover crop, but can also be harvested for profit, and may be a solution to the lack of cover crop adoption. Pennycress (*Thlaspi arvense* L.) is a winter annual oilseed currently under development that may fit the need for a cash cover crop in the upper Midwest. Pennycress can provide a host of ecosystem services, including early season pollinator foraging resources [31,32], weed suppression [33], erosion reduction [34], and the reduction of N leaching. When relay-cropped with soybean, pennycress significantly reduced N concentration in both the soil and leachate to a depth of 60 cm, compared with a mono-cropped soybean summer annual tilled system [35–37]. Unlike many other cover crops, pennycress can provide farmers with direct economic returns through the harvest of its oil rich seeds, which contain 24–39% wt/wt oil [38,39]. High erucic acid and linoleic acid content (33 and 22 wt/wt%) makes the oil an excellent candidate for uses as a biofuel, an industrial lubricant, and a component of biodegradable plastics [32,38,40–42]. Recent mutational breeding efforts have discovered pennycress strains with an oil profile comparable to canola, opening the market to products for human consumption [43]. Additionally, after seeds have been pressed for oil, the seed meal can then be used as high-protein animal feed, fuel, and as a biofumigant depending on the oil and glucosinolate profiles [33,38,40,41].

Previous work on incorporating pennycress into cropping systems has focused on the effect of pennycress on the following crops, with little attention on the effect of the crops preceding pennycress [33,37,44,45]. Additionally, as a developing crop, agronomic best management practices such as adequate fertilizer regimens and seeding methods have yet to be established [45]. Only one study thus far has quantified the effects of fertilizer on pennycress yield and oil quality, and no significant difference was found between fertilizer rates [46]. Similarly, only one study has examined the effect of drilling vs. broadcast seeding pennycress seeds. Phippen et al. [47] found that drilled plots outperformed broadcast planted plots regardless of the seeding rate. However, a study by Carr [48] postulated that broadcast seeding would be preferable as pennycress is positively photoblastic [49]. The objectives of this study were to: (1) determine the effect of seeding method and sweet corn N application on pennycress yield; and (2) analyze the effect of pennycress and N rate on residual inorganic soil N following sweet corn.

## 2. Materials and Methods

### 2.1. Site Description

Field studies were conducted at the Southern Research and Outreach Center in Waseca, MN, USA, and at the Rosemount Research and Outreach Center in Rosemount, MN, USA from 2017 to 2019. The Waseca plots (44°04′31″ N 93°31′31″ W) were located on a Webster/Nicollet clay loam soil (fine-loamy, mixed, superactive, mesic Typic Endoaquolls and fine-loamy, mixed, superactive, mesic Aquic Hapludolls, respectively), and the Rosemount plots (44°42′25″ N 93°04′22″ W) were located on a Waukegan silt loam (fine-silty over sandy or sandy-skeletal, mixed, superactive, mesic Typic Hapludolls). Spring wheat (*Triticum aestivum* L.) was grown the summer prior to all studies.

### 2.2. Weather Data

Precipitation and air temperature data were obtained from the NOAA reporting weather station at Waseca and Rosemount, respectively, and departures from the 30-year average (1981–2010) temperature and accumulated precipitation [50] were calculated. Air temperatures during pennycress spring growth were colder than the 30-year average in all environments (Table 1). Precipitation during pennycress establishment in September was 45–46% lower than average at both locations in 2017–2018 (−49.8 mm and −42.1 mm for Rosemount and Waseca, respectively), and 70% and 186% higher than average at both locations in 2018–2019 (+65.0 mm and +174.1 mm; Table 1). In May, when pennycress was maturing, precipitation was high across all environments (average +42.8 mm), as well as June in 2017–2018 (+34.5 mm and +27.3 mm for Rosemount and Waseca, respectively; Table 1).

**Table 1.** Monthly climate data and departures from the 30-year average for Rosemount, MN over the 2017–2018 and 2018–2019 growing seasons.

| Month | 2017–2018 † | | | | 2018–2019 † | | | |
|---|---|---|---|---|---|---|---|---|
| | Mean Air Temperature (°C) | Departure from Average ‡ (°C) | Accumulated Precipitation (mm) | Departure from Average ‡ (mm) | Mean air Temperature (°C) | Departure from Average ‡ (°C) | Accumulated Precipitation (mm) | Departure from Average ‡ (mm) |
| | | | | **Rosemount** | | | | |
| Jun. | 20.5 | 0.9 | 91.4 | −28.5 | 21.5 | 1.9 | 154.4 | 34.5 |
| Jul. | 22.3 | 0.4 | 138.7 | 24.4 | 22.0 | 0.1 | 111.0 | −3.3 |
| Aug. | 18.9 | −1.8 | 128.8 | 8.7 | 21.1 | 0.4 | 102.1 | −18.0 |
| Sep. | 17.9 | 1.9 | 42.4 | −49.8 | 17.4 | 1.4 | 157.2 | 65.0 |
| Oct. | 9.6 | 0.7 | 98.6 | 26.0 | 6.2 | −2.7 | 90.9 | 18.3 |
| Nov. | −0.6 | −0.7 | 1.8 | −51.5 | −3.3 | −3.4 | 37.6 | −15.7 |
| Dec. | −8.2 | 0.0 | 8.4 | −22.6 | −5.0 | 3.2 | 47.2 | 16.3 |
| Jan. | −11.1 | −0.4 | 24.9 | −1.5 | −11.0 | −0.4 | 35.1 | 8.6 |
| Feb. | −11.9 | −4.2 | 28.2 | 5.1 | −13.2 | −5.5 | 72.8 | 49.6 |
| Mar. | −1.4 | −0.9 | 23.1 | − 35.3 | −4.3 | −3.8 | 59.2 | 0.8 |
| Apr. | 1.0 | −6.9 | 50.3 | −23.9 | 6.5 | −1.4 | 129.5 | 55.4 |
| May. | 18.6 | 4.3 | 108.7 | 6 .1 | 11.6 | −2.6 | 173.0 | 70.3 |
| Jun. | 21.5 | 1.9 | 154.4 | 34.5 | 20.0 | 0.4 | 119.8 | −0.1 |
| | | | | **Waseca** | | | | |
| Jun. | 21.1 | 0.9 | 105.6 | −14.0 | 21.5 | 1.2 | 146.9 | 27.3 |
| Jul. | 23.1 | 0.9 | 166.7 | 54.0 | 21.7 | −0.5 | 111.1 | −1.6 |
| Aug. | 19.1 | −1.9 | 99.3 | −21.8 | 20.7 | −0.3 | 121.7 | 0.6 |
| Sep. | 17.7 | 1.4 | 51.5 | −42.1 | 17.8 | 1.5 | 267.7 | 174.1 |
| Oct. | 9.8 | 0.8 | 105.1 | 37.0 | 6.4 | −2.6 | 80.4 | 12.3 |
| Nov. | −0.4 | −0.8 | 4.4 | −50.7 | −4.2 | −4.6 | 34.2 | −20.9 |
| Dec. | −8.4 | −0.5 | 22.9 | −14.8 | −5.1 | 2.8 | 53.3 | 15.6 |
| Jan. | −11.7 | −1.3 | 46.9 | 15.0 | −11.2 | −0.7 | 32.5 | 0.6 |
| Feb. | −11.7 | −4.2 | 29.5 | 4.0 | −14.1 | −6.6 | 77.0 | 51.5 |
| Mar. | −1.6 | −1.2 | 29.6 | −33.9 | −4.2 | −3.7 | 51.0 | −12.5 |
| Apr. | 0.6 | −7.2 | 89.4 | 7.5 | 6.9 | −1.0 | 108.0 | 26.1 |
| May. | 18.5 | 3.6 | 134.2 | 34.0 | 12.0 | −2.8 | 161.1 | 60.9 |
| Jun. | 21.5 | 1.2 | 146.9 | 27.3 | 20.2 | 0.0 | 84.7 | −34.9 |

† Mean air temperature and accumulated precipitation were recorded from the NOAA reporting weather station at the Rosemount Research and Outreach Center, Rosemount, MN, USA and the Southern Research and Outreach Center, Waseca, MN USA. ‡ Calculated departure from the 30-year average (1981–2010) temperature and accumulated precipitation using data collected at the Rosemount or Southern Research and Outreach Centers. The average frost-free period was April 7–October 29 in 2017, April 20–October 18 in 2018 and April 28–October 25 in 2019.

### 2.3. Experimental Setup

The study was arranged in a randomized complete block design, with four replications of 5 N fertilizer rates and three cover crop treatments. Sweet corn variety "GSS 1477" was planted mid-June of 2017 and 2018 with a six-row planter (John Deere 7100 MaxEmerge, Deere and Co., Moline, IL, USA) at a rate of 60,540 seeds ha$^{-1}$ in 4.6 × 9 m plots at a depth of 3.8 cm with 76 cm row spacing. Based on soil tests, additional P and K were not required. Nitrogen fertilizer treatments were surface applied pre-plant as urea. The five N treatments were: 0 (control), 65, 135, 135 as split applications of 67 (denoted as 135s) and 200 kg N ha$^{-1}$. The 135s treatment had 67 kg N ha$^{-1}$ applied pre-plant, and the remaining 67 kg N ha$^{-1}$ was applied between the V4 and V6 growth stages. Representative sweet corn subsamples were harvested by hand for fresh market yield at the end of August.

Corn stalks were cut and left in the field using a stalk chopper. The three-cover crop treatments were established mid-September, following sweet corn harvest. Cover crop treatments ('Cover Crop') were no cover crop (fallow control), compared to pennycress established by two seeding methods: direct broadcast + light incorporation, termed 'DBC+INC' (Avenger high-clearance tractor, LeeAgra Inc., Lubbock, TX, USA, using an orbital air seeder, Gandy Co., Owatonna, MN, USA and custom made disturbance units of spring-tines followed by a log chain drag; (Figure 1), compared to no-till drilling 15 cm rows at a depth of 0.3 cm termed 'DRILL' (Interseeder Technologies, Woodward, PA, USA).

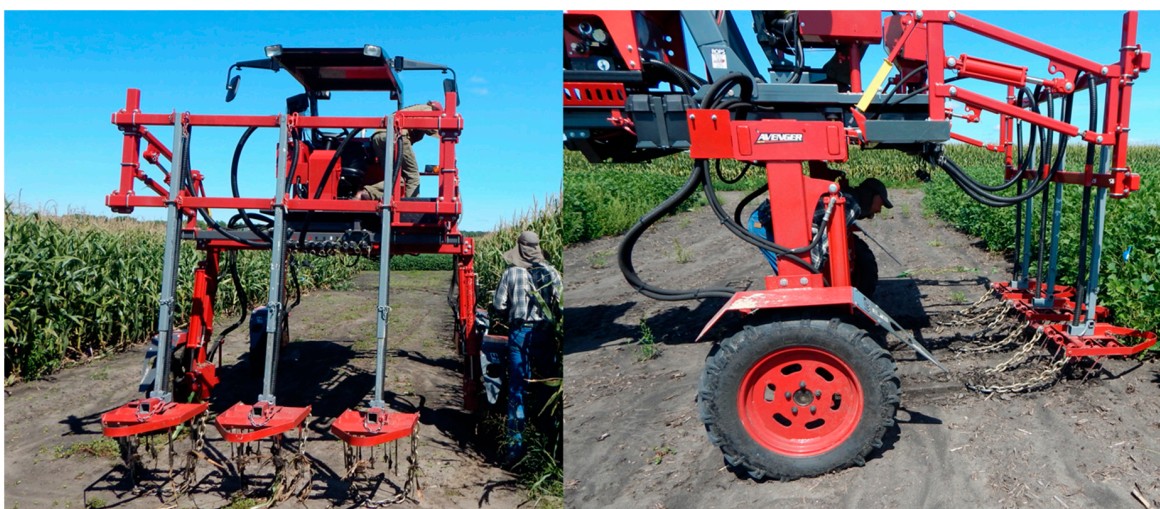

**Figure 1.** Modified Avenger high-clearance tractor outfitted with a Gandy Co. Orbit Air Seeder. Drop stanchions support cover crop seed delivery housing and soil incorporation units. Photo credit: Eric Ristau.

### 2.4. Sampling and Analyses

Pennycress was hand-harvested at maturity in mid-June the following year from a 0.5 m$^2$ quadrat and dried to constant weight in a 35 °C oven. Dry grain was threshed for using a stationary low-profile plot thresher (Almaco, Nevada, IA, USA) in 2016 and 2017, and a stationary Wintersteiger LD 350 thresher (Wintersteiger Inc., Sal Lake City, UT) in 2018. Baseline soil samples (Table 2) were collected prior to sweet corn planting on a site-wide basis from 0- to 60-cm depth, and analyzed for organic matter (OM) [51]; bulk density [52]; cation exchange capacity (CEC) [53]; pH [54]; calcium (Ca), magnesium (Mg), potassium (K), and phosphorus (P) [55,56]; as well as total inorganic ('mineral') N ($N_{min}$) which is the sum of ammonium-N ($NH_4^+$-N) [57] and nitrate-N ($NO_3^-$-N) [58,59]. Soil $N_{min}$ at sweet corn planting was 217 kg and 65 kg, and 146 kg and 89 kg N ha$^{-1}$ at Rosemount 2017 and 2018, and Waseca 2017 and 2018, respectively. Soil samples were also taken in each plot from 0–60 cm at pennycress seeding and harvest and analyzed for $N_{min}$ content [60]. Aboveground biomass was hand harvested from a 0.5 m$^2$ quadrat in each pennycress plot when pennycress reached physiological

maturity in early-June, dried to constant weight at 95 °C, and ground before analysis with a CN analyzer (vario Max cube, Elementar, Langenselbold, Germany) to obtain percent N content. Percent N was multiplied by aboveground biomass weight to obtain N uptake.

**Table 2.** Pre-plant soil characteristics of Rosemount and Waseca field sites in 2017 and 2018 from 0–60 cm.

| Environment | W/V | CEC | pH | OM | Ca | Mg | K | P | $N_{min}$ |
|---|---|---|---|---|---|---|---|---|---|
| | g cm$^{-3}$ | cmol kg$^{-1}$ | | | | mg kg$^{-1}$ | | | kg ha$^{-1}$ |
| Rosemount | | | | | | | | | |
| 2017 | 1.5 | 26.5 | 5.9 | 4.7 | 2889.8 | 586.2 | 194.0 | 23.8 | 217 |
| 2018 | 1.5 | 16.7 | 6.1 | 3.6 | 1744.1 | 427.3 | 96.1 | 8.5 | 65 |
| Waseca | | | | | | | | | |
| 2017 | 1.5 | 24.0 | 6.2 | 3.9 | 2504.2 | 548.7 | 212.5 | 16.0 | 146 |
| 2018 | 1.4 | 36.9 | 6.3 | 5.3 | 4068.3 | 643.9 | 164.5 | 8.5 | 89 |

OM = organic matter, W/V = bulk density, CEC = cation exchange capacity, CA = calcium, Mg = magnesium, K = potassium, P = phosphorus, $N_{min}$ = inorganic N ($NH_4^+$-N + $NO_3^-$-N).

Statistical analyses were performed using the MIXED procedure in the statistical software SAS (SAS Institute Inc., Cary, NC, USA). A combined analysis was prevented due to the significant ($p \leq 0.05$) year by location interaction for all parameters. Fixed effects were N treatment, cover crop treatment and their interactions for residual soil $N_{min}$ at pennycress harvest, pennycress yield, pennycress biomass, and pennycress N uptake (sequestration). For residual soil $N_{min}$ at sowing, the sowing method treatment cover crop treatments had not yet been seeded, so only N treatment was considered a fixed effect. For pennycress yields, only DRILL and DBC+INC treatments within cover crop treatment were considered. Random effects were block nested within year by location, and corresponding interactions with fixed effects. To meet ANOVA assumptions of homoscedastic variance, residual inorganic soil N data was natural log transformed for analysis and back transformed for presentation. Means for all response variables were separated using Fisher's LSD at $p \leq 0.05$.

## 3. Results

### 3.1. Sweet Corn

Average yields in fresh weight of unhusked ears of sweet corn were 15.6 and 16.4 Mg ha$^{-1}$ at Rosemount in 2017 and 2018, respectively, and were 13.2 and 15.6 Mg ha$^{-1}$ at Waseca in 2017 and 2018, respectively (not including the 0 N controls). The sweet corn yields were within averages for the surrounding county at each location [61].

### 3.2. Soil N

#### 3.2.1. N Treatment

Nitrogen treatment in sweet corn affected residual soil $N_{min}$ at the time pennycress was planted across all environments ($p \leq 0.05$; Table 3). Increasing rates of N applied to sweet corn increased the level of residual $N_{min}$ at pennycress seeding (Table 4). The 200 N treatment consistently had the highest residual $N_{min}$. The 0 and 65 N treatments consistently had the lowest residual $N_{min}$ and were no different from each other. On average, the 200 N treatment had 38–80% more residual $N_{min}$ than the 0 and 65 N treatments (Table 4).

**Table 3.** Results (*p*-values) of the mixed-model analysis of variance of residual soil inorganic N ($N_{min}$) at pennycress seeding and harvest, pennycress grain yield, biomass and N uptake.

| Environment | Fixed Effects | Residual Soil $N_{min}$ | | Pennycress † | | |
| | | Seeding | Harvest | Yield | Biomass | N Uptake |
|---|---|---|---|---|---|---|
| Rosemount 2017 | N | <0.001 | <0.001 | 0.625 | 0.634 | 0.678 |
| | Cover Crop | - | <0.001 | 0.834 | 0.574 | 0.396 |
| | N × Cover Crop | - | 0.022 | 0.978 | 0.837 | 0.844 |
| Rosemount 2018 | N | <0.001 | 0.718 | 0.277 | 0.627 | 0.665 |
| | Cover crop | - | <0.001 | 0.073 | 0.490 | 0.299 |
| | N × Cover crop | - | 0.360 | 0.766 | 0.987 | 0.864 |
| Waseca 2017 | N | <0.001 | 0.116 | <0.001 | 0.057 | 0.047 |
| | Cover crop | - | <0.001 | 0.572 | 0.105 | 0.097 |
| | N × Cover crop | - | 0.537 | 0.225 | 0.808 | 0.590 |
| Waseca 2018 | N | <0.001 | 0.197 | 0.265 | 0.909 | 0.780 |
| | Cover crop | - | 0.295 | 0.791 | 0.474 | 0.961 |
| | N × Cover crop | - | 0.211 | 0.430 | 0.706 | 0.611 |

† The fallow cover crop control data is not included in pennycress data analysis.

At the time of seeding pennycress, residual soil $N_{min}$ in the 135 N and 135s N treatments following sweet corn harvest did not differ in three of the four environments, with the exception being Waseca 2018, where the split application resulted in a higher residual $N_{min}$ compared to a single at-seeding application (Table 4). Additionally, in two of the four environments, residual $N_{min}$ from split applications of N totaling 135 kg N ha$^{-1}$ (135s N) did not differ compared to applying 200 kg N ha$^{-1}$ (200 N), while residual $N_{min}$ remaining from a single at-seeding application of 135 kg N ha$^{-1}$ (135 N) was always less than that remaining after applying 200 kg N ha$^{-1}$ (200 N).

By the time pennycress was harvested, nitrogen treatment had no effect on residual soil $N_{min}$ in three of four environments (Table 3). At Rosemount in 2017, N treatment had an interactive effect with cover crop treatment on residual soil $N_{min}$ at pennycress harvest (Figure 2). Control plots with no cover crop left higher residual N than in plots seeded to pennycress at all N rates except 0 kg N ha$^{-1}$.

**Table 4.** Residual soil inorganic N ($N_{min}$) by sweet corn N treatment at pennycress seeding at Rosemount and Waseca, MN in 2017 and 2018 from 0–60 cm.

| N Treatment | Rosemount † | | | | Waseca † | | | |
| | 2017 | | 2018 | | 2017 | | 2018 | |
| | (kg ha$^{-1}$) | | | | | | | |
|---|---|---|---|---|---|---|---|---|
| 0 | 28.0 | C | 33.8 | C | 33.8 | C | 52.9 | C |
| 65 | 33.8 | C | 47.9 | C | 42.1 | BC | 54.6 | BC |
| 135 | 61.1 | B | 85.1 | B | 45.6 | B | 62.4 | B |
| 135s | 73.1 | AB | 77.3 | B | 42.4 | BC | 79.7 | A |
| 200 | 88.6 | A | 172.1 | A | 70.9 | A | 85.1 | A |

† Values within columns followed by different letters are significantly different ($p \leq 0.05$). Data were log transformed for analysis and back transformed for presentation. Abbreviations: 135s, 135 kg N ha$^{-1}$ split application.

### 3.2.2. Cover Crop Treatment

Seeding a cover crop reduced residual soil $N_{min}$ remaining by pennycress harvest in Rosemount 2018 and Waseca 2017 (Table 3). For these site-years, leaving plots fallow left on average 27% to 42% more residual $N_{min}$ compared to seeding a pennycress cover crop, regardless of the method of seeding used (Table 5); this trend was not seen in Waseca in 2018. Across all environments, the method of seeding pennycress did not affect residual soil $N_{min}$ remaining at pennycress harvest (Table 5, Figure 2).

**Table 5.** Residual inorganic soil N ($N_{min}$) at pennycress harvest across cover crop treatments at Rosemount and Waseca, MN in 2017 and 2018 from 0–60 cm.

| Cover Crop Treatment | Rosemount | | | | Waseca | | | |
|---|---|---|---|---|---|---|---|---|
| | 2017 | | 2018 | | 2017 | | 2018 | |
| | ($kg\ ha^{-1}$) | | | | | | | |
| No Pennycress | 45.6 A [†] | | 55.1 | A | 78.4 | A | 41.0 | A |
| DBC+INC | 31.2 A | | 32.2 | B | 58.1 | B | 38.8 | A |
| DRILL | 30.2 A | | 32.1 | B | 56.9 | B | 41.4 | A |

[†] Values within a column followed by different letters are significantly different ($p \le 0.05$). Data were log transformed for analysis and back transformed for presentation. Abbreviations: DBC+INC, Directed Broadcast with Incorporation; DRILL, No-till grain drill.

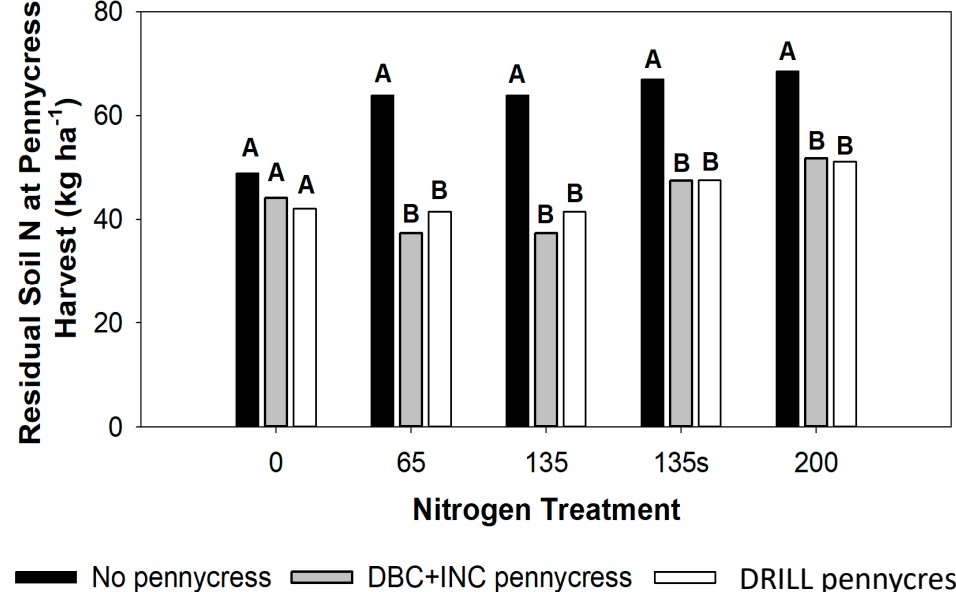

**Figure 2.** Interactive effect of sweet corn N fertilizer rate (N Treatment) and cover crop treatment on mean residual inorganic soil N at pennycress harvest for the Rosemount in 2017. Values within N treatment followed by different letters are significant ($p \le 0.05$). Data was log transformed for analysis and back transformed for presentation.

*3.3. Pennycress Grain Yield, Biomass Production and Nitrogen Uptake*

Pennycress grain yield was not affected by the seeding method (Table 3), and was affected by N treatment in only one of four site-years, Waseca in 2017, (Table 3) where higher N rates applied to sweet corn tended to result in higher pennycress yields (Table 6). Pennycress yields were low, averaging 601 and 387 kg $ha^{-1}$ at Rosemount and 324 and 568 kg $ha^{-1}$ at Waseca in 2017 and 2018, respectively (Table 6). The pennycress seeding method did not affect aboveground pennycress biomass (Table 3), which ranged from 1164 to 1454 kg $ha^{-1}$ at Rosemount and 1019 to 1718 kg $ha^{-1}$ at Waseca in 2018 and 2019, respectively. Nitrogen uptake by pennycress was the highest for the 200 N treatment at Waseca in 2017, but did not differ at the other three environments (Tables 3 and 6). The average N uptake for N treatments across years and locations ranged from 15 to 28 kg N $ha^{-1}$.

**Table 6.** Pennycress seed yield and N Uptake as impacted by sweet corn N treatment for Rosemount and Waseca during the 2017 and 2018 growing seasons.

| N Treatment | Rosemount | | | | Waseca | | | |
| --- | --- | --- | --- | --- | --- | --- | --- | --- |
| | Seed Yield | | N Uptake | | Seed Yield | | N Uptake | |
| | 2017 | 2018 | 2017 | 2018 | 2017 † | 2018 | 2017 † | 2018 |
| | kg ha$^{-1}$ | | | | | | | |
| 0 | 666 | 287 | 22 | 15 | 329 BC | 546 | 18 B | 21 |
| 65 | 553 | 380 | 23 | 20 | 260 D | 487 | 17 B | 17 |
| 135 | 631 | 381 | 28 | 18 | 345 AB | 581 | 18 B | 21 |
| 135s | 594 | 442 | 28 | 20 | 266 CD | 681 | 18 B | 21 |
| 200 | 643 | 420 | 28 | 20 | 397 A | 559 | 22 A | 24 |

† Values within columns followed by different letters are significantly different ($p \leq 0.05$). Abbreviations: 135s, 135 kg N ha$^{-1}$ split application.

## 4. Discussion

Sweet corn yields in our experiment were within average for the respective counties for the given year, as well as within average for the past five years. While weeds were persistent, regular manual removal kept weed pressure low. Our results are consistent with previous studies that show increasing fertilizer rates led to increased levels of residual soil N post-sweet corn harvest [6]. Previous research has shown the capability of pennycress to reduce residual inorganic soil N [35,37,62], and our study corroborated these results with the exception of Waseca in 2018, where there was no difference between pennycress treatments and bare ground. Waseca received 23% and 34% higher than average rainfall in May and June of 2018, respectively, causing extremely wet field conditions (Table 1). As a result, we were unable to apply pre-emergent herbicides in the spring, resulting in high weed pressure. While weeds are undesirable in an agricultural setting, they are a form of living cover, and have been shown to reduce residual inorganic soil N up to 39 kg ha$^{-1}$ [63], and are likely the cause of the lack of difference between pennycress and bare ground plots in Waseca 2018.

Rukavina et al. [46] found that maximum pennycress seed production of 888 kg ha$^{-1}$ occurred with 56 kg N ha$^{-1}$ applied to pennycress as a split application of 28 kg N ha$^{-1}$ in both the fall and spring. In our study, residual soil N$_{min}$ in the fall after sweet corn harvest and before pennycress seeding ranged from 28 to 172 kg N ha$^{-1}$, equal to or far exceeding the fall application in Rukavina's study. However, our yields ranged from 601 to 287 kg ha$^{-1}$, lower than Rukavina et al. [46] (Table 6). The main cause of low pennycress yields in our study was seed loss due to shatter from heavy rains at maturity, however it is likely that spring fertilization, as in Rukavina's study, may be beneficial to yields.

Additionally, this study found no difference in yield or biomass of pennycress based on seeding method (i.e., Brillion vs. Broadcasted and Incorporated). Drilling seed is more expensive, largely based on the greater time requirement compared to broadcasting [64,65], so the ability to broadcast pennycress seed without affecting yield improves the economic feasibility of cropping systems including pennycress. Noland et al. [16] and Phippen et al. [47] also found that pennycress biomass and seed yield did not differ between drilled and broadcast seeding, corroborating the results of our study.

Noland et al. [16] also looked at pennycress N sequestration, but sampled plants in early spring, before peak biomass, and found only 11.4 kg N ha$^{-1}$, comparable to the 13.4 kg N ha$^{-1}$ sequestered by pennycress in our study in early spring (averaged across all environments) [66]. Weyers et al. [35] found that pennycress sequestered an additional 35 to 40 kg N ha$^{-1}$ of soil N by late-spring compared to bare ground control plots—more than 40% greater than observed. Unlike our study, however, Weyers et al. [35] applied N fertilizer directly to pennycress in the spring, which may have increased N availability and pennycress N uptake. For comparison to pennycress, cereal rye (*Secale cereale* L.), the most popular winter annual in the upper Midwest, has been shown to sequester around 2 to 114 kg N ha$^{-1}$ in the northern USA [67–69]. On the other hand, Dean and Weil [70] showed the

Brassicaceae species forage radish (*Raphanus sativus* L. cv. Daikon), oilseed radish (*Raphanus sativus* L. cv. Adagio) and rape (*Brassisca napus* L. cv. Dwarf Essex) on average sequestered 32 kg more N ha$^{-1}$ than rye, though their study was carried out in the Mid-Atlantic USA, which experiences a longer growing season than the Midwest. This previous research shows that pennycress and other oilseeds can compete with rye in amount of N sequestered, however, our results suggest there are optimal field conditions under which this may occur. Future replicated field studies that grow rye and pennycress side by side would help elucidate key environmental factors to N sequestration in pennycress and how it will perform against the current most popular winter cover. While winter rye has value as a winter cover and as a forage, in the upper Midwest it does not reach grain maturity until August [71], leaving no time for a summer crop. Alternatively, as pennycress matures in late June, it offers farmers an option to harvest their cover as a second cash crop, creating a financial incentive to plant a winter cover crop.

In this study, the amount of N pennycress sequestered did not increase with increased availability of soil N at seeding in three of four environments. Blackshaw et al. [72] found a different response in which pennycress shoot biomass and percent tissue N increased with increasing N fertilizer rates. However, the minimum fertilizer treatment in the Blackshaw et al. [72] experiment was 40 mg N kg$^{-1}$ (approximately 90 kg N ha$^{-1}$), and increased up to 240 mg N kg$^{-1}$ (approximately 540 kg N ha$^{-1}$) (soil depth: 0–15 cm, soil type: Typic Haplustoll sandy loam), 270% greater than the highest N treatment used in our study or that would be found under best management practices in the field. This may mean pennycress requires extremely concentrated levels of soil N in order to increase sequestration. Additionally, Blackshaw et al. [72] experiments were carried out in pots in a greenhouse environment. There are many factors in the field that are not present in a greenhouse setting that could have limited the ability of pennycress to sequester N, such as N availability in the soil profile, timing of rain events or drought, and root structure.

While this study did not directly measure nitrogen lost to leaching or gaseous emissions, it is important to consider these losses when analyzing the system as a whole. In general, sweet corn utilizes only 20–42% of applied nitrogen [6], with the rest susceptible to environmental loss. Pennycress sequestered 11.6% to 78.6% of residual $N_{min}$ at pennycress harvest (Tables 4 and 6), with the percentage sequestered increasing as the amount of residual $N_{min}$ decreased, further suggesting the presence of a limiting factor to pennycress sequestration in the field. The remaining $N_{min}$ was subject to environmental loss. However, as residual $N_{min}$ values at pennycress harvest exceeded those at pennycress planting (Table 5), further mineralization of nitrogen must also be considered. To achieve a more complete view of the nitrogen cycling in a sweet corn–pennycress rotation, the use of resin bags as in Carlson [73] would be useful.

When drawing conclusions from these results, it must be considered that pennycress yields were low in all environments, which may have minimized response to N treatments (Table 6). Heavy rains before pennycress harvest in all site-years exacerbated seed lost to shatter, and may be the primary reason seed yields were so low. The pennycress variety 'MN106' has the potential to yield upwards of 2,000 kg ha$^{-1}$ [37,45], but in our trial yields ranged from 260 to 671 kg ha$^{-1}$. As a wild accession, MN106 still exhibits weedy characteristics. Seed shatter from preharvest dehiscing of silicles is one of the more troublesome weedy traits [38], as it can result in seed losses of up to 300 times the initial seeding rate [73]. In one study by Cubins, pennycress seed lost to shatter accounted for up to 70% of the total yield [45]. Breeders at the University of Minnesota are working to select traits that prevent shattering, which will reduce yield losses in future lines [74].

## 5. Conclusions

The rate of nitrogen (N) applied to sweet corn did not affect the following pennycress seed yield (with the exception of Waseca in 2017), total aboveground biomass or biomass N content, indicating that maximum pennycress yields may be attained under relatively low fertilizer regimes in most years. Higher fertilizer N applications to sweet corn resulted in higher residual $N_{min}$ at pennycress seeding, but these differences did not carry through to the following summer regardless of the presence or

absence of pennycress. Environmental losses to leaching, runoff, volatilization or denitrification were not monitored in this study. However, seeding pennycress after sweet corn did reduce residual $N_{min}$ in three of four site years compared to leaving the ground fallow. While there may be a limit to the amount of N pennycress can sequester compared to cover crops like rye, it still served as an effective N catch crop. These findings support the use of the least costly method of seeding, DBC+INC, which did not jeopardize pennycress yields. However, further research is needed to verify this finding across a range of environments, as drilled seedings historically outperform broadcast seedings in a variety of crops. Pennycress can be successfully grown utilizing only the residual inorganic N from a preceding sweet corn crop. As a winter annual cash cover crop, pennycress has the potential to follow sweet corn. However, further research is needed to maximize pennycress yield and environmental benefits before this system can be recommended to farmers.

**Author Contributions:** Conceptualization, M.S.W. and C.J.R.; methodology, S.A.M., M.S.W. and C.J.R.; validation, S.A.M. and M.S.W.; formal analysis, S.A.M. and M.S.W.; investigation, S.A.M. and M.S.W.; resources, M.S.W.; data curation, M.S.W. and S.A.M.; writing—original draft preparation, S.A.M.; writing—review and editing, S.A.M., M.L.W., M.S.W., R.W.G. and R.L.B.; visualization, S.A.M., M.S.W. and M.L.W.; supervision, M.S.W.; project administration, M.S.W.; and funding acquisition, M.S.W. All authors have read and agree to the published version of the manuscript.

**Funding:** This research was funded by University of Minnesota Forever Green Agricultural Initiative.

**Conflicts of Interest:** The authors declare no conflict of interest. The funders had no role in the design of the study; in the collection, analyses, or interpretation of data; in the writing of the manuscript; or in the decision to publish the results.

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
