# Peer review of "Pennycress as a Cash Cover-Crop: Improving the Sustainability of Sweet Corn Production Systems"

_agronomy, doi:10.3390/agronomy10050614_

Round 1

Reviewer 1 Report

The paper reports the results of a two year field experiment at two sites investigating different N fertilizer levels and pennycress cultivation as cover crop following sweet corn on residual inorganic soil nitrogen and pennycress yields. The major contributions are that pennycress yields were not affected by different fertilizer levels although inorganic soil nitrogen prior to its cultivation increased with higher N fertilizer applications to sweet corn, and that pennycress partly reduced inorganic soil nitrogen at pennycress harvest compared with a fallow treatment.

The paper is generally well written and reports interesting results on the use of a cash cover crop to reduce inorganic N burdens, which are potential contaminants of surface and ground water and are particularly high after sweet corn cultivation. The introduction is well structured and provides a good background to the objectives. Material and methods would improve with subtitles (site description, experimental set-up, sampling and analyses or similar) and slight restructuring according to suggested subtitles. Also the results would benefit from a restructuring (the two first sub-chapters can be combined to Residual soil inorganic N and the third and fourth sub-chapter to pennycress yield and N uptake). The description should focus on the major treatment effects, which are later on discussed. The split N application does not have clear effects on remaining inorganic N and can be addressed less pronounced. Also the results section should only contain a description of the results, no interpretation or discussion! The discussion should be restructured according to the objectives and it is too superficial, often the cited references are just mentioned to confirm the current results, but more details would be necessary to understand how the results are compared. The use of cover crops is attractive as it can sequester N, which would be otherwise lost during winter. The comparison of residual N before and after pennycress harvest, in addition with N uptake is missing. How much N was possibly lost with and without pennycress cultivation? I think this is a major aspect of this study. Regarding this aspect, the yield of sweet corn at different N levels would be interesting, as for pennycress lowest N level already led to maximum yields, which means higher N application rates only lead to higher losses. Thus, pennycress might not have sufficient N uptake potential to substantially reduce N losses after sweet corn, where usually high residual N is present.

Line 93 and 94: the abbreviations SROC and RROC are not necessary as later on the two sites are referred to as Waseca and Rosemount

Line 99 and following: climatic conditions are described here, Table 2 and 3 should move here, although it is questionable if the whole tables are necessary here or could be added to supplementary material, as for my understanding it is not really needed for discussion of results, maybe a summary of these tables would be enough

Line 106: abbreviations (DBC+INC and BRIL) should be explained at first mention

Line 109: Was only urea applied or did you also apply other mineral fertilizers for P and K supply for instance?

Line 116: It is not mentioned when pennycress was seeded.

Line 119: Were both treatments sown in 15 cm rows at 0.3 cm depths? If yes, how could the broadcast treatment be sown in precise rows?

Line 125-126: Although for each analytical method a reference is mentioned, it would be helpful to have at least a short description of the method used. As the used method has an influence on the comparability of results, the reader would be forced to look at all the references.

Line 128: Please change the sentence to the following for better understanding. Soil Nmin at sweet corn planting was 217 and 65, and 146 and 89 kg N ha-1 at Rosemount in 2017 and 2018, and Waseca in 2017 and 2018, respectively.

Lines 130-134: Sounds similar to line 120

Line 141: the term site year is confusing

Line 147: “homogeneous” variances?

Line 151: The weather data belong to material and methods, they are not results from the applied treatments but describe the conditions during the experiment

Line 174: Number of table is not correct, please change it to 4, and keep the same order of the measured parameters in table title as in the table

Line 176: The term planting method in the table is misleading as the major statistical difference is found between no pennycress and the two cover crop treatments.

Line 179: The term “site years” is very confusing

Line 183: The sentence starting with …, likely due to … is an interpretation and does not belong to the results, where results should only be described

Line 185: double-crop treatment is confusing, better use “cover crop treatment”

Line 186: “control treatment” instead of check plots?

Line 187: Table 5, “Residual soil inorganic N averaged by sweetcorn N treatment…”

Line 188: What does MN mean?

Line 190 and 191: “Cover crop treatment”

Line 196: Sentence starting with “, showing …” is an interpretation not a description, and belongs to discussion.

Line 198: Re-number table (Table 6)

Line 202: “Cover crop treatment”

Line 200: the font size of axes is huge, and should be reduced

Line 210, 212 and 214: sentences starting with “, however… “ and “, indicating…” are again an interpretation!

Line 217: “a subset of double crop treatment” is confusing, maybe can be removed?

Line 222: It would be interesting to compare the uptake with the difference of residual N min before and after pennycress cropping, to see how much of the available N min was sequestered in pennycress and how much still lost

Line 223: Table 7, order of parameters in title should be similar in table

Line 224: Table 8

Line 237: remove “compared with bare ground” as bare ground did not sequester any N, and add “in” after “much higher than”

Line 238: The lower N uptake in current study compared to Weyers might result from a low N availability in spring when pennycress requires more nutrients. Here it would be interesting to compare the N sequestered by pennycress with available N after sweet corn harvest and after pennycress harvest. It might show that a considerable amount of N was lost during winter, leading to a low availability in spring. It is often a problem of cover crops that their N requirement before winter is not as high as residual N sources in the soil, leading to N losses during winter.

Line 239: This argument is not valid. N concentration was likely lowered at later harvesting time due to a dilution effect, but the total biomass should have been increased. N taken up by the crop is usually not lost, maybe translocated to the seeds. N uptake is lower at younger development stages.

Line 242-245: it is not clear where this argument should lead. This comparison seems to be unnecessary.

Line 247: please introduce nitrogen as N at first mention and then consequently use N throughout the whole document

Line 247-259: A reason for the missing response might be losses of N during winter time as mentioned before!

Line 267: When the used seeding methods are different from the usual drilling and broadcast, it should be explained more detailed in materials and methods!

Line 269: drilling seed is more expensive … than what?

Line 274: “oilseed species”

Line 285: Did the heavy rains before harvest occur in both years at both sites?

Line 300: So where did the N go? This should be discussed.

Author Response

Agronomy – 759072 Editor Comments and Authors Reply

R1, Comments:

The paper reports the results of a two year field experiment at two sites investigating different N fertilizer levels and pennycress cultivation as cover crop following sweet corn on residual inorganic soil nitrogen and pennycress yields. The major contributions are that pennycress yields were not affected by different fertilizer levels although inorganic soil nitrogen prior to its cultivation increased with higher N fertilizer applications to sweet corn, and that pennycress partly reduced inorganic soil nitrogen at pennycress harvest compared with a fallow treatment.

The paper is generally well written and reports interesting results on the use of a cash cover crop to reduce inorganic N burdens, which are potential contaminants of surface and ground water and are particularly high after sweet corn cultivation. The introduction is well structured and provides a good background to the objectives.

  1. Material and methods would improve with subtitles (site description, experimental set-up, sampling and analyses or similar) and slight restructuring according to suggested subtitles.
    1. We agree with your suggestion, and added the requested details.
  2. Also the results would benefit from a restructuring (the two first sub-chapters can be combined to Residual soil inorganic N and the third and fourth sub-chapter to pennycress yield and N uptake). The description should focus on the major treatment effects, which are later on discussed. The split N application does not have clear effects on remaining inorganic N and can be addressed less pronounced.
    1. Thank you for the suggestions. We agreed and edited the manuscript for clarity.
  3. Also the results section should only contain a description of the results, no interpretation or discussion!
    1. We agreed and lines pertaining to interpretation were removed and relocated to the discussion section as appropriate.
  4. The discussion should be restructured according to the objectives and it is too superficial, often the cited references are just mentioned to confirm the current results, but more details would be necessary to understand how the results are compared.
    1. We agreed and improved the discussion based on your suggestions.

The use of cover crops is attractive as it can sequester N, which would be otherwise lost during winter.

  1. The comparison of residual N before and after pennycress harvest, in addition with N uptake is missing. How much N was possibly lost with and without pennycress cultivation? I think this is a major aspect of this study.
    1. We recognize there are limitations with this study, and even though we do not have the point estimates you are referring to we do know how well the pennycress performed relative to no pennycress. In most cases the N services provided by pennycress were small, with exception of one of the environments. We think much of this is related to your suspicions that the N losses moved beyond pennycress rooting depth. Since we experienced wet and cold springs for both years it is likely that N was leached prior to being sequestered by the pennycress. Even with the potential unaccounted for leaching, pennycress did reduce soil residual N at one of the environments.
  2. Regarding this aspect, the yield of sweet corn at different N levels would be interesting, as for pennycress lowest N level already led to maximum yields, which means higher N application rates only lead to higher losses. Thus, pennycress might not have sufficient N uptake potential to substantially reduce N losses after sweet corn, where usually high residual N is present.
    1. You bring up a good point, and we have learned in another study that the N requirement for pennycress is quite low. Our data suggest that oilseed yield and quality is achieved at 40 to 50 kg N applied per ha.
  3. Line 93 and 94: the abbreviations SROC and RROC are not necessary as later on the two sites are referred to as Waseca and Rosemount
    1. These were removed and we ref. to Wascea and Rosemount throughout the paper.
  4. Line 99 and following: climatic conditions are described here, Table 2 and 3 should move here, although it is questionable if the whole tables are necessary here or could be added to supplementary material, as for my understanding it is not really needed for discussion of results, maybe a summary of these tables would be enough
    1. We agree and make the following suggestions. We decided to combine Table 2 and 3 into a landscape table. The table numbers were off and we corrected them globally. Table 1 is the weather/climate table now.
  5. Line 106: abbreviations (DBC+INC and DRILL) should be explained at first mention
    1. We agree, and edited the manuscript to reflect your suggestions.
  6. Line 109: Was only urea applied or did you also apply other mineral fertilizers for P and K supply for instance?
    1. No. P and K were not required based on soil tests. We edited the manuscript addressing your question.
  7. Line 116: It is not mentioned when pennycress was seeded.
    1. We added the date to the text.
  8. Line 119: Were both treatments sown in 15 cm rows at 0.3 cm depths? If yes, how could the broadcast treatment be sown in precise rows?
    1. We agree and make the following changes: the depth and rows are meant for the drilled treatments, and the wording has been rearranged to convey this.
  9. Line 125-126: Although for each analytical method a reference is mentioned, it would be helpful to have at least a short description of the method used. As the used method has an influence on the comparability of results, the reader would be forced to look at all the references.
    1. We agreed and provided references to soil test methods.
  10. Line 128: Please change the sentence to the following for better understanding. Soil Nmin at sweet corn planting was 217 and 65, and 146 and 89 kg N ha-1 at Rosemount in 2017 and 2018, and Waseca in 2017 and 2018, respectively.
    1. We agreed and made the suggested changes to the manuscript.
  11. Lines 130-134: Sounds similar to line 120
  12. Line 141: the term site year is confusing
  13. Line 147: “homogeneous” variances?
    1. Yes, homoscedastic variance. We edited to reflect your suggestion.
  14. Line 151: The weather data belong to material and methods, they are not results from the applied treatments but describe the conditions during the experiment
    1. We agreed and made the following changes: moved the weather section to materials and methods.
  15. Line 174: Number of table is not correct, please change it to 4, and keep the same order of the measured parameters in table title as in the table
    1. We apologize for this oversight. We combined the weather/climate tables and updated all table numbers. We also edited the table captions to felect
  16. Line 176: The term planting method in the table is misleading as the major statistical difference is found between no pennycress and the two cover crop treatments.
    1. We agree that the non-pennycress check is driving much of the observed N differences. However, since pennycress is a small seeded crop, planting methodology impacts on biomass and yield are important findings.
  17. Line 179: The term “site years” is very confusing.
    1. Site-year is commonly used across agronomic domains, however, so is environment. We agreed and changed site year to environment.
  18. Line 183: The sentence starting with …, likely due to … is an interpretation and does not belong to the results, where results should only be described
    1. We agreed and moved to the discussion as suggested.
  19. Line 185: double-crop treatment is confusing, better use “cover crop treatment”
    1. We agreed that double-crop is confusing and at times a bit misleading. We opted to change double-crop treatment to planting method.
  20. Line 186: “control treatment” instead of check plots?
    1. Both work for us. We agreed and edited based on your suggestion.
  21. Line 187: Table 5, “Residual soil inorganic N averaged by sweetcorn N treatment…”
    1. The line numbers are not lining up. We are using the documents that were submitted and reviewed. We think the reviewer is referring to and suggesting a change here. We are more than willing to accommodate their suggestions, we need direction on these issues.
  22. Line 188: What does MN mean?
    1. Minnesota. We added the abbreviation in the M&M.
  23. Line 190 and 191: “Cover crop treatment”
    1. We agreed and edited based on your suggestions.
  24. Line 196: Sentence starting with “, showing …” is an interpretation not a description, and belongs to discussion.
    1. We agree and made the following changes: phrase was moved to discussion.
  25. Line 198: Re-number table (Table 6).
    1. All table numbers have been updated
  26. Line 202: “Cover crop treatment”
    1. We agreed and made the suggested changes.
  27. Line 200: the font size of axes is huge, and should be reduced
    1. We agreed and made the suggested changes.
  28. Line 210, 212 and 214: sentences starting with “, however… “ and “, indicating…” are again an interpretation!
    1. We agreed and edited in the spirit of results and moved the interpretation to the discussion.
  29. Line 217: “a subset of double crop treatment” is confusing, maybe can be removed?
  30. Line 222: It would be interesting to compare the uptake with the difference of residual N min before and after pennycress cropping, to see how much of the available N min was sequestered in pennycress and how much still lost
    1. We did explore that scenario, and did provide the data in Table 5. The planting method main effects show that pennycress on average is impacting residual N when compared to no pennycress. The issue is that based on our data the service to extractable inorganic soil N is small most years. 
  31. Line 223: Table 7, order of parameters in title should be similar in table
    1. We agreed and made the suggested changes.
  32. Line 224: Table 8
    1. All table numbers have been updated.
  33. Line 237: remove “compared with bare ground” as bare ground did not sequester any N, and add “in” after “much higher than”
    1. We agreed and edited for clarity.
  34. Line 238: The lower N uptake in current study compared to Weyers might result from a low N availability in spring when pennycress requires more nutrients. Here it would be interesting to compare the N sequestered by pennycress with available N after sweet corn harvest and after pennycress harvest. It might show that a considerable amount of N was lost during winter, leading to a low availability in spring. It is often a problem of cover crops that their N requirement before winter is not as high as residual N sources in the soil, leading to N losses during winter
    1. We do agree that fall or early N losses do occur since small cover crop plants provide small services. However, the bulk of N loses in the upper Midwest takes place when the tile lines are running from Apr and May. During this time pennycress is well positioned to sequester available N. Line numbers were not lining up with comments.
  35. Line 239: This argument is not valid. N concentration was likely lowered at later harvesting time due to a dilution effect, but the total biomass should have been increased. N taken up by the crop is usually not lost, maybe translocated to the seeds. N uptake is lower at younger development stages.
    1. We agreed with the dilution effect can mask N single. We collected data for both physiological and harvest maturity, but chose to present the harvest maturity data since soil N data only corresponded at harvest.    
  36. Line 242-245: it is not clear where this argument should lead. This comparison seems to be unnecessary.
    1. We agreed that the argument is unclear as currently presented. The comparison is misleading. The comparison illustrates the cover crop N sequestration services of pennycress using the metric of soil residual N. We added text to alleviate the confusion.
  37. Line 247: please introduce nitrogen as N at first mention and then consequently use N throughout the whole document
    1. We agreed and fixed it globally.
  38. Line 247-259: A reason for the missing response might be losses of N during winter time as mentioned before!
    1. See comment reply #38. We do agree that fall or early N losses do occur since small cover crop plants provide small services. However, the bulk of N loses in the upper Midwest takes place when the tile lines are running from Apr and May. During this time pennycress is well positioned to sequester available N. Line numbers were not lining up with comments.
  39. Line 267: When the used seeding methods are different from the usual drilling and broadcast, it should be explained more detailed in materials and methods!
    1. We agreed and removed the statement that our methods differed from the standard definitions of drilled and broadcasted with incorporation.
  40. Line 269: drilling seed is more expensive … than what?
    1. We agreed as presented leaves one a bit confused. Drills are the gold standard, however, drills, even big drills are still smaller than largest air seeders. Size isn't the only factor, broadcasting is faster on average. Time and labor cost are greater for the drill than broadcasting. We edited the manuscript for clarity.
  41. Line 274: “oilseed species”
    1. We agreed, and made the suggested change.
  42. Line 285: Did the heavy rains before harvest occur in both years at both sites?
    1. Yes. We updated the manuscript to reflect your comment.
  43. Line 300: So where did the N go? This should be discussed.
    1. We do have an incomplete picture of the N balance. We recognize the limitations of our N picture. Even though we cannot account for all the N, we did gain insight to the N needed for pennycress. It is likely that the wet springs moved much or the N beyond the pennycress rooting depth, however, the pennycress did assimilate and remove N from the environment, especially in 2017 at Waseca.

Reviewer 2 Report

Dear authors:

IMHO the paper needs a deep revision. I have some comments that can help to enhance the article.

The approach of this article is inappropriate due to the aim of sowing cover crops is for enhancing soil qualities and avoid losses in the main crop and in this article looks like that the main crop is pennycress, not sweet corn.

Revise all tables and figures numbers

  1. - Title and keywords should not have the same words
  2. - Revise and complete all Latin names
  3. - Line 17: Provide units
  4. - Line 19-20: Explain better what do you want to say.
  5. - Line 45: Change the term “Shoulder season” is not a technical term. Revise the whole document and use technical terms.
  6. - Line 53-65: Local for the US, what happened in the rest of the world with cover crops? Rest of America, Europe, Asia…

Cite these articles in the introduction: “Identification of Arable Marginal Lands under Rainfed Conditions for Bioenergy Purposes in Spain” and “Tall wheatgrass (Thinopyrum ponticum (Podp)) in a real farm context, a sustainable perennial alternative to rye (Secale cereale L.) cultivation in marginal lands”

  1. - Line 111: Harvest by hand? You cannot extrapolate results for farmers
  2. - Line 123-124: Provide international rules used for taking soil samples.
  3. - Line 151-164: Specify the annual free frost period. “Weather data” should be Material and Methods
  4. - Line 174: Table 1 again? Units? Revise all.
  5. - Line 210: “Unclear”? Be more straightforward. There is not a technical word. Revise the whole document there are lots of these terms, such as line 237: “Much higher”…Line 263: “Much lower”
  6. - Change planting for sowing in the document. You plant a tree and you sow seeds.
  7. - Line 251: “So it is possible” Be more specific and clarify the sentence.
  8. - Line 308-311: How can you say that you recommend to farmers this method if you do not corroborate the agronomical study with the economic study.
  9. – Around half of the references are not citations for journals with impact factor.

Author Response

Agronomy – 759072 Editor Comments and Authors Rebuttal

R2; Comments:

IMHO the paper needs a deep revision. I have some comments that can help to enhance the article

  1. The approach of this article is inappropriate due to the aim of sowing cover crops is for enhancing soil qualities and avoid losses in the main crop and in this article looks like that the main crop is pennycress, not sweet corn.
    1. We do agree with the idea of a cash cover crop like pennycress is novel. Traditionally, cover crops outside of forage production systems were primarily used for ecosystem services. While the initial purpose of utilizing a cover crop is to improve environmental outcomes, we propose the use of oilseed cash cover crops that provide the potential of economic return while maintaining ecosystem services.  You are also correct that sweetcorn per se, isn't the primary focus of this work, however, improving sweetcorn sustainability with the use of a cash cover crop is. While extensive research has been conducted on optimizing sweet corn systems, pennycress is still a new crop, hence the necessity of focusing this research on pennycress.
  2. Revise all tables and figures numbers
    1. We agreed and updated all figure and table numbering.
  3. Title and keywords should not have the same words
    1. We agreed and removed the similar words.
  4. Revise and complete all Latin names.
    1. We agreed and fixed it globally.
  5. Line 17: Provide units
    1. We agreed and provided the units.
  6. Line 19-20: Explain better what do you want to say.
    1. We understand that preparing a clean, and readable manuscript is paramount. We will make any changes suggested. However, we require more direction to address your concerns.
  7. Line 45: Change the term “Shoulder season” is not a technical term. Revise the whole document and use technical terms.
    1. We agreed and removed “shoulder season” and stated the specific months where summer annual crop production is not possible due to winter.
  8. Line 53-65: Local for the US, what happened in the rest of the world with cover crops? Rest of America, Europe, Asia…
    1. We understand that Agronomy has a broad international readership, and we do as much to expand our inference sphere to encompass as many regions as possible. With that being said, there are limits to the scope and applicability of the systems outlined in the manuscript. We believe this manuscript addresses challenges of cover crop integration throughout the Upper Midwest. While this manuscript offers new insight into cash cover crop integration into sweetcorn systems, it is not a review of cover crop status across the globe. If your journal is looking for a special issue review on cover crops, my team is more than willing to provide leadership.
  9. Cite these articles in the introduction: “Identification of Arable Marginal Lands under Rainfed Conditions for Bioenergy Purposes in Spain” and “Tall wheatgrass (Thinopyrum ponticum (Podp)) in a real farm context, a sustainable perennial alternative to rye (Secale cereale L.) cultivation in marginal lands”
    1. See comment 55. We felt that the addition of these papers does not support the unique challenges associated with the upper Midwest.
  10. Line 111: Harvest by hand? You cannot extrapolate results for farmers
    1. We understand your concerns around the scalability of results, and as applied scientists, we do everything in our power to reflect real-world conditions. Even though we hand-harvested, our subsamples accurately represented the experimental unit (i.e., the plot). We added text indicating to the reader that sub-samples represent the experimental units.
  11. Line 123-124: Provide international rules used for taking soil samples.
    1. We were unaware that there are international rules for soil sampling. We followed the SSSA guidelines.
  12. Line 151-164: Specify the annual free frost period. “Weather data” should be Material and Methods
    1. We agreed and added the requested information.
  13. Line 174: Table 1 again? Units? Revise all.
    1. We agreed and combined the tables.
  14. Line 210: “Unclear”? Be more straightforward. There is not a technical word. Revise the whole document there are lots of these terms, such as line 237: “Much higher”…Line 263: “Much lower”
    1. We agreed and fixed it globally.
  15. Change planting for sowing in the document. You plant a tree and you sow seeds.
    1. We also plant crops. We chose to leave the word plant in play.
  16. Line 251: “So it is possible” Be more specific and clarify the sentence.
    1. We agreed and edited the manuscript based on your suggestions.
  17. Line 308-311: How can you say that you recommend to farmers this method if you do not corroborate the agronomical study with the economic study.
    1. We agree that more research including economics are needed before farmers can be advised to adopt. We edited the manuscript to better reflect that point.
  18. Around half of the references are not citations for journals with impact factor.
    1. Not really sure how to address this. We counted over 48 from relevant journals.

Reviewer 3 Report

The authors have presented a well organized study in an important research area. The paper is well written without any small mistakes. My only concern is some of the data that are not shown in the paper (described below). It would be better if authors include that data or give a better explanation for not showing that data.

I suggest you report Rosemount 2017 data in the Table 3. I understand that you report there is significant interaction bw N treatment and planting treatment. If you don't show these data, it's very difficult to see if there is an effect on catching N with Pennycress.

I suggest same for the Table 4 data

Please include remaining data related to Table 5. YOu have included 2017 Weseca. But you don't show data for 2018 Weseca, 2017 and 2018 Rosemount data.

Could you please explain why did you analyze N content at harvest maturity instead of late-spring?

Author Response

Agronomy – 759072 Editor Comments and Authors Rebuttal

R3; Comments:

The authors have presented a well-organized study in an important research area. The paper is well written without any small mistakes. My only concern is some of the data that are not shown in the paper (described below).

  1. It would be better if the authors include that data or give a better explanation for not showing that data.
    1. We agreed and opted to show all data.
  2. I suggest you report Rosemount 2017 data in Table 3. I understand that you report there is significant interaction between N treatment and planting treatment. If you don't show these data, it's very difficult to see if there is an effect on catching N with Pennycress
    1. We agree this is confusion. The data for Rosemount is reported. What is not reported are the means of separation letters. To clear up the confusion we removed the table and move values inline.
  3. I suggest the same for Table 4 data
    1. We agreed and combined tables added data, and move other tables to inline.
  4. Please include the remaining data related to Table 5. Y0u have included 2017 Waseca. But you don't show data for 2018 Waseca, 2017 and 2018 Rosemount data.
    1. We agreed the table layout is a bit confusing. To deal with the confusion we added the N impacts for Rosemount 2017 and 2018 and Waseca 2018.
  5. Could you please explain why did you analyze N content at harvest maturity instead of late-spring?
    1. We were interested in total plant N at physiological maturity and harvest maturity. However, we only have soil N data from harvest maturity, thus we chose to present the harvest maturity data.

Round 2

Reviewer 1 Report

The manuscript has considerably improved. I still have a few comments, some of them content-related. Particularly the discussion is in large parts very descriptive and the reasons for findings are not discussed deep enough. I think the paper would benefit, if possible reasons for low N uptake (N losses) and lack of effect of sowing method are discussed more deeply.

Sowing or seeding method might be better than planting method?

I hope the Line numbers are clear.

Line 127: as well as in June 2017-2018

Line 210: A combined analysis …

Line 395: This sentence is not clear to me: “In addition to no planting method impacts on pennycress biomass performance, pennycress N uptake was not affected by sweetcorn N treatment across all environments”. Could you please rephrase it, so that the meaning becomes clear.

Line 449: … with the exception of Waseca in 2018, where there was no difference between pennycress treatments and bare ground. The addition would make the paragraph a bit clearer.

Line 455: When citing in the text, usually the year is added in parentheses or is it different in this Journal? A study by Weyers et al. (YEAR)?

Line 457: This sentence is not clear to me with the changes. What do you mean with plants luxury spend on N? And I think something is missing in the sentence “it is likely that the applied N fertilizer Weyers et al., directly to pennycress… ?

Line 455-456: I am still not sure, why you mention rye here. What do you exactly want to tell with this comparison. As a grain of course it is very different from Brassica species. The range shows that it can sequester much more N under certain conditions. So I would think, ok then why don’t you grow rye here… If you want to keep this comparison, it would be interesting to have some more information on the benefits and drawback of rye as cover crop. The question arising here is why do you promote pennycress cultivation if rye has a much higher N uptake potential? I am sure there are reasons, but it is not clear yet.

Line 588: … greater time requirement to seed the same about of hectares

Line 590-595: These comparisons can be condensed and should be complemented by a discussion of the reasons why in some studies there are effects of seeding method and in others there are no effects. Just to mention some found effects others not is just a description and does not give any new insights of the effect.

Line 630: the Environmental losses should be mentioned or discussed in the discussion more deeply, as this could be a reason for the lacking effect of residual soil N on yields and N uptake. I think this is not clear enough currently. It is somehow mentioned in Lines 553-555, but it would be nice to have references here indicating how much N can be lost via leaching and gaseous emissions during the winter…

Line 632: … compared to sweetcorn followed by fallow …

Line 635: Planting method as a factor within planting method treatment did not affect

Line 640-643: You mentioned that pennycress yields were very low, so is the conclusion that pennycress can be successfully grown really valid? To really conclude this it would be good to know about the economical benefit. Was there really an economic benefit even with low yields? Maybe this cannot be fully answered in this study and maybe pennycress has the potential of a good cash cover crop. However from this study with low yields and low N sequestration, I am not fully convinced that pennycress is such a good solution.

Author Response

Reviewer 1:

The manuscript has considerably improved. I still have a few comments, some of them content-related. Particularly the discussion is in large parts very descriptive and the reasons for findings are not discussed deep enough. 

  1. I think the paper would benefit, if possible reasons for low N uptake (N losses) and lack of effect of sowing method are discussed more deeply.
    1. We agreed and environmental N loss discussed as in comment 12.
  2. Sowing or seeding method might be better than planting method?
    1. The plant and agricultural sciences commonly interchange “planting” with “sowing or seeding.” We used planting for crops that were planted such as the sweet corn and soybean and seeding for pennycress. In addition, we agreed that “Planting Method” was not the best way to describe the cover crop planting methods and the fallow check. We opted to replace “Planting Method” with “Cover Crop Treatment”.
  3. Line 127: as well as in June 2017-2018
    1. We agreed and added dated for clarification
  4. Line 210: A combined analysis …
    1. We did not perform a combine analysis. What is shown in Figure 1, are the interaction effects of N x CC. Mean separation were assigned to the marginal means associated with N treatment. A combined analysis implies that a pooled error term was used.
  5. Line 395: This sentence is not clear to me: “In addition to no planting method impacts on pennycress biomass performance, pennycress N uptake was not affected by sweetcorn N treatment across all environments”. Could you please rephrase it, so that the meaning becomes clear.
    1. We agreed and reworded for clarity.
  6. Line 449: … with the exception of Waseca in 2018, where there was no difference between pennycress treatments and bare ground. The addition would make the paragraph a bit clearer.
    1. We agreed and thank you for this suggestion. The wording “where there was no difference between pennycress treatments and bare ground” has been added to the sentence to add clarity.
  7. Line 455: When citing in the text, usually the year is added in parentheses or is it different in this Journal? A study by Weyers et al. (YEAR)?
    1. We agreed and thank you for catching this. Actually, the correct format for this journal is to use the author's name or author et al. followed by the reference number [x]. This has been changed throughout the text.
  8. Line 457: This sentence is not clear to me with the changes. What do you mean with plants luxury spend on N? And I think something is missing in the sentence “it is likely that the applied N fertilizer Weyers et al., directly to pennycress… ?
    1. We agreed and edited the manuscript for clarity.
  9. Line 455-456: I am still not sure, why you mention rye here. What do you exactly want to tell with this comparison. As a grain of course it is very different from Brassica species. The range shows that it can sequester much more N under certain conditions. So I would think, ok then why don’t you grow rye here… If you want to keep this comparison, it would be interesting to have some more information on the benefits and drawback of rye as cover crop. The question arising here is why do you promote pennycress cultivation if rye has a much higher N uptake potential? I am sure there are reasons, but it is not clear yet.
    1. As now explained (P12 bottom), rye is added here as a comparison to pennycress. Rye is the most popular cover crop (also mentioned in this section), but cover cropping is still slow to being adopted by farmers, as explained in the Intro. Pennycress is being developed as a “cash cover crop” (also explained in the Intro) as a way to incentivize grower adoption without government subsidies. Therefore, for practical reasons, it makes good sense to compare pennycress to rye, as well as other Brassica species (e.g., Dean and Weil 2009). The authors wish to keep this comparison in the text.
  10. Line 588: … greater time requirement to seed the same about of hectares
    1. We agreed and reworded as suggested.
  11. Line 590-595: These comparisons can be condensed and should be complemented by a discussion of the reasons why in some studies there are effects of seeding method and in others there are no effects. Just to mention some found effects others not is just a description and does not give any new insights of the effect
    1. We agreed and found there was an error in our discussion of the literature. The confounding sentences have been removed and remaining sentences reworded for clarity.
  12. Line 630: the Environmental losses should be mentioned or discussed in the discussion more deeply, as this could be a reason for the lacking effect of residual soil N on yields and N uptake. I think this is not clear enough currently. It is somehow mentioned in Lines 553-555, but it would be nice to have references here indicating how much N can be lost via leaching and gaseous emissions during the winter
    We agreed and added a paragraph in the discussion.
  13. Line 632: … compared to sweetcorn followed by fallow …
    1. We agreed and made the suggested change.
  14. Line 635: Planting method as a factor within planting method treatment did not affect
    1. We reworded as suggested.
  15. Line 640-643: You mentioned that pennycress yields were very low, so is the conclusion that pennycress can be successfully grown really valid? To really conclude this it would be good to know about the economical benefit. Was there really an economic benefit even with low yields? Maybe this cannot be fully answered in this study and maybe pennycress has the potential of a good cash cover crop. However from this study with low yields and low N sequestration, I am not fully convinced that pennycress is such a good solution.
    1. We agreed to the importance of economics and how said economics drives adoption. With that being said, and unfortunately, Economics is beyond the scope of this particular study. Presently, markets for pennycress are still in their infancy. Furthermore, pennycress is still in the process of being fully domesticated by crop breeders, and still has a way to go before being as genetically refined or as consistent as rapeseed. The objectives were to evaluate how pennycress responds to different levels of sweet corn fertility, and whether it significantly reduces residue nitrate N.

Reviewer 2 Report

Dear authors,

thank you for your efforts, the manuscript improved the quality after revision. But there are comments suggested that you did not take into account.

Please revise again all Latin names.

Revise the introduction. We do not need a review but we need a general vision around the world, the manuscript is very local.

Revise discussion and conclusions, it is the weakest part of the manuscript.

Author Response

Reviewer 2

Dear authors,

Thank you for your efforts, the manuscript improved the quality after revision. But there are comments suggested that you did not take into account.

  1. Please revise again all Latin names.

Thank you. All Latin names were modified in the manuscript again.

  1. Revise the introduction. We do not need a review but we need a general vision around the world, the manuscript is very local.

Thank you. According to the instructions for this journal, in the introduction section, “The current state of the research field should be reviewed carefully and key publications cited.” This suggests that a review is appropriate, otherwise the current state of research in this field would not be known. However, we understand the need to provide a broader overview. We added some language to the beginning of the introduction that discusses regions where sweet corn is grown.

  1. Revise discussion and conclusions, it is the weakest part of the manuscript.

Thank you for the comment. Unfortunately, it is very vague and we are unclear as to what needs to be fixed. However, Reviewer 1, provided directed feedback that addresses your comment.